# Development and Evaluation of Matrices Composed of β-cyclodextrin and Biodegradable Polyesters in the Controlled Delivery of Pindolol

**DOI:** 10.3390/pharmaceutics12060500

**Published:** 2020-05-30

**Authors:** Agnieszka Lis-Cieplak, Filip Charuk, Marcin Sobczak, Anna Zgadzaj, Agata Drobniewska, Łukasz Szeleszczuk, Ewa Oledzka

**Affiliations:** 1Department of Biomaterials Chemistry, Chair of Analytical Chemistry and Biomaterials, Medical University of Warsaw, Faculty of Pharmacy, Banacha 1, 02-097 Warsaw, Poland; alis@wum.edu.pl (A.L.-C.); filip.charuk01@gmail.com (F.C.); marcin.sobczak@wp.pl (M.S.); 2Department of Environmental Health Sciences, Medical University of Warsaw, Faculty of Pharmacy, Banacha 1, 02-097 Warsaw, Poland; azgadzaj@wum.edu.pl (A.Z.); agata.drobniewska@wum.edu.pl (A.D.); 3Department of Physical Chemistry, Chair of Physical Pharmacy and Bioanalysis, Medical University of Warsaw, Faculty of Pharmacy, Banacha 1, 02-097 Warsaw, Poland; lszeleszczuk@wum.edu.pl

**Keywords:** β-cyclodextrin, natural initiator, polymeric conjugate, pindolol, functionalization of polymers, polyesters, drug delivery systems

## Abstract

Polymer-drug conjugates are currently being more widely investigated for the treatment of hypertension. In view of the above, in the first stage of our work, we used nontoxic β-cyclodextrin (β-CD) as effective, simple, inexpensive, and safe for the human body initiator for the synthesis of biocompatible and biodegradable functionalized polymers suitable for the medical and pharmaceutical applications. The obtained polymeric products were synthesized through a ring-opening polymerization (ROP) of ε-caprolactone (CL), d,l-, and l,l-lactide (LA and LLA). The chemical structures of synthesized materials were elucidated based on ^1^H NMR and solid-state carbon-13 cross-polarization/magic angle spinning nuclear magnetic resonance (^13^C CP/MAS NMR) analysis, while the incorporation of β-CD molecule into the polymer chain was confirmed by matrix-assisted laser desorption/ionization time-of-flight mass spectrometry (MALDI-TOF MS). Furthermore, molecular modeling has been applied to investigate the intrachain rigidities and chain architectures for several representative structures. The obtained and thoroughly characterized branched matrices were then used to generate the first β-cyclodextrin/biodegradable polymer/β-blocker conjugate through the successful conjugation of pindolol. The conjugates were fabricated by carbodiimide-mediated coupling reaction. The branched biodegradable materials released the drug in vitro in a sustained manner and without “burst release” and thus have the ability to treat different heart diseases.

## 1. Introduction

Arterial hypertension is a disease of humanity and the primary cause of the population’s premature death [1]. If this condition is neglected or improperly treated, it increases the risk of serious complications such as heart failure, ischaemic and haemorrhagic stroke, and chronic kidney disease [2]. The ultimate goal of antihypertensive treatment is to reduce the death rate and cardiovascular morbidity. Clinical outcome studies indicate that lowering blood pressure with various classes of antihypertensives, including angiotensin-converting enzyme inhibitors (ACEIs), angiotensin receptor blockers (ARBs), β-blockers (BBs), Ca channel blockers (CCBs), and thiazide-type diuretics will all reduce hypertension complications. All of these drugs are suitable for the initiation and maintenance of antihypertensive treatment either as monotherapy or in some combination with each other. All these drugs, alone or in combination, display a reasonable hypertensive effect and tolerability in hypertensive patients, and substantial evidence is also obtained to prevent the occurrence of cerebrovascular and cardiovascular disease [3].

One of the main advances in modern pharmacy is the evaluation of new drugs, new drug types, or new drug delivery systems (DDSs) that allow the delivery of an active substance at the right time with minimal side effects and with the most preferred method of administration at a particular place. Nevertheless, it is exceedingly difficult to synthesize such innovative drugs/systems.

Development of a new drug is not a guarantee of its launch. Today, only 1–2 from approximately 10,000 new substances passes through all phases of clinical trials and can then be marketed. In fact, its long-term effects on the human body is unclear in the case of new drugs.

Modern pharmacy is also engaged in improving the pharmacokinetics of existing medicines in addition to the development of new drugs by modifying their medicinal formulations, such as the synthesis of macromolecular therapeutic systems or polymer controlled release systems, including the synthesis the macromolecular conjugates of antihypertensive drugs [4]. Synthetic and natural polymers are frequently used for this purpose. Biodegradable and/or bioresorbable polyesters form the largest and most promising biomaterial class. This is attested by their widespread use in different medical and pharmaceutical applications, such as surgery or implantology [5,6]. These polymers are optimally suited to stay in the body only as long as they serve their function and then are disappearing without the need of a second surgical intervention [7,8].

In recent years, a special attention has been given to natural and safe-for-the-human-body compounds for use as initiators or catalysts of the ring opening polymerization (ROP) of cyclic esters. The ROP of these monomers is a convenient and efficient method to obtain biodegradable or bioresorbable polyester matrices. Among the variety of ROP processes, including the well-established polymerization using metal-based initiators or catalysts [9], the application of natural enzymes [10], amino acids [11], or organic acids [12] is particularly advantageous in view of medicine and pharmacy. To the group of natural ROP initiators belongs cyclodextrins (CDs)-cyclic oligosaccharides and seminatural products obtained from starch by an enzymatic conversion. CDs consisting of 6, 7, and 8 glucose units are named α-, β-, and γ-CD, respectively. Due to their ability to form host–guest inclusion complexes by entrapping small molecules (guests) in the hydrophobic cavity of the macrocyclic sugar (host), they have many practical applications as drug delivery carriers in the pharmaceutical field [13,14,15,16]. This ability leads also to specific effects on the small molecules of cyclic esters-monomers during ROP by temporary complexation of the monomer [17]. Furthermore, CDs are capable of initiating ROP of cyclic esters. The molecular mechanism of δ-valerolactone (VL)/β-CD reactant complex formation for the initiation reaction of the ROP of VL was analysed by the molecular simulations [18]. The authors confirmed the formation of the reactant complex structure. The comparison between the single and multiple β-CD models revealed that the formation was more frequent and that the distance for the nucleophilic attack was shorter in the multiple model [19]. Importantly, in the innovatory studies of Harada and coworkers [19,20,21,22]. the use of α-, β- and γ-CDs as the ROP initiators of various lactones, e.g., ε-caprolactone (CL), VL, and β-butyrolactone (β-BL) in mild conditions (100 °C), was reported. The yield of the polymerization products depended on the lactone structure and the cyclodextrin cavity size. The authors demonstrated that the inclusion of lactones in the CD cavity and secondary hydroxyl groups of the CD molecule plays an important role in the initiation of polymerization. CDs were found to form inclusion complexes at 100 °C without water [19]. However, the yield of the polymerization product in the reactions initiated by β-CD (especially poly(ε-caprolactone) (PCL)) obtained in their studies was only about 15%, with the product average molecular weight of 1700 Da. Harada and coworkers elucidated this occurrence by a low activity of CL because of the low Lewis basicity of the carbonyl oxygen of CL [20]. Moreover, the authors claimed that, in the ROP of l,l-lactide (LLA), the order of the poly(l,l-lactide) (PLLA) yields with CDs initiators was γ-CD > β-CD > α-CD. In addition, CDs initiated ROP of d,l-lactide (LA) more efficiently than this of LLA, demonstrating chiral selectivity for the monomer.

Even if Harada’s and coworkers’ proposed approach indicates the possibility of the ideal polymerization method [19], it would be reasonable to explore these reactions in greater depth. According to the literature, the melting point of the LLA monomer is around 96 °C, so its low reactivity at 100 °C could be expected. Zinck and coworkers [23] demonstrated that substantial to quantitative conversions of LLA were achieved for ROP temperatures between 120 and 150 °C, whereas a temperature of 180 °C resulted in polymer degradation, characterized by a deep brown color. Nonetheless, the monomer conversion was approximately 1% after 24 h, when the LLA ROP was performed at 100 °C.

Given the above considerations, in the first step of our study, we investigated the ROP of CL, LLA, and LA in the presence of nontoxic β-CD initiator in bulk at again 100 as well as 140 °C. Such monomers were selected because of their many desirable properties for medical applications as biodegradation to nontoxic products and biocompatibility with tissues. Besides, the combination of these monomers with β-CD exhibited ineffective polymerization results in Harada and coworkers’ studies. The obtained polymers were thoroughly characterized using ^1^H NMR, solid-state carbon-13 cross-polarization/magic angle spinning nuclear magnetic resonance (^13^C CP/MAS NMR), size exclusion chromatography with multi-angle light scattering (SEC-MALLS), matrix-assisted laser desorption/ionization time-of-flight mass spectrometry (MALDI-TOF MS), and thermogravimetric (TGA) analysis as well as biological tests. Additionally, molecular modeling has been applied to investigate the intrachain rigidities and chain architectures for several representative structures.

In the next step of our study, synthesized and characterized branched polyester materials have been employed as the carriers for covalent conjugation of pindolol (PDL) for local delivery. Pindolol (1-(1H-indol-4-yloxy)-3-(isopropylamino)-2-propanol) (marketed products: Visken^®^, Visken Quinze, Apo-Pindol, Apo-Pindolol) is a synthetic β-adrenergic receptor blocking agent with intrinsic sympathomimetic activity [24]. It is an effective agent for treating hypertension in pregnancy, a disease that complicates up to 5% of all pregnancies. PDL is a nonselective β-adrenergic antagonist (β-blocker) which possesses intrinsic sympathomimetic activity (ISA) in therapeutic dosage ranges but does not possess quinidine-like membrane-stabilizing activity. PDL causes no changes in utero or in umbilico-placental vascular impedance or blood flow, has no effect on fetal haemodynamics, and does not affect fetal cardiac function. PDL is well absorbed in the gastrointestinal (GI) tract, presenting 90% bioavailability and a plasma half-life of 3–4 h, which makes it a candidate for a sustained release formulation [25].

Recent investigations have been carried out on the complexation of PDL with CDs with the aim of improving the in vitro transcorneal permeability of the drug in eye drop formulations [26]. Furthermore, the fluorimetric analysis of the interaction between PDL and different CDs in order to determine the apparent stability constants of the complexes and the thermodynamic parameters associated to complexation have been investigated [27]. Corrected excitation and emission wavelengths in different solvents have been reported, and the effect of solvent on the Stokes shifts of these compounds has been analysed using the Lippert equation. In addition, the sorption of PDL and other compounds on a crosslinked polymer containing β-CD has been studied by Zornoza and coworkers [28]. The similar rapid kinetic profiles obtained for the four solutes was related to the high swelling capacity of the polymer, which enables the expansion of its network and the diffusion process. The differences detected in the sorption capacities of the solutes were associated to their ability to form inclusion complexes within the CD cavities. The sorption isosteric heats determined evidence the heterogeneity of the polymer, with the CD cavity being the most favourable adsorption site.

Nevertheless, according to the literature and our knowledge, branched conjugates of PDL constructed from biodegradable polyester chains and β-CD, which form the central core, have not yet been obtained and described. In this sense, the rate of drug release from the synthesized conjugates has been analyzed in vitro.

## 2. Materials and Methods

### 2.1. Materials

β-Cyclodextrin (β-CD) of pharmaceutical grade was purchased from CTD, Inc. (Alachua, Florida, USA) and constantly dried before use. l,l-lactide (*cis*-3,6-dimethyl-1,4-dioxane-2,5-dione, LLA, 98.0%, Aldrich Co., Poznan, Poland) and d,l-lactide (3,6-dimethyl-1,4-dioxane-2,5-dione, LA, 98.0%, Aldrich Co.) were recrystallized from dried ethyl acetate in a dry nitrogen atmosphere and then thoroughly dried in a vacuum before use. ε-Caprolactone (2-oxepanone, 99%, CL, Sigma-Aldrich Co.) was dried with calcium hydride and distilled under argon atmosphere before use. Pindolol (PDL, 1-(1H-indol-4-yloxy)-3-(isopropylamino)-2-propanol, solid, ≥98%, Sigma-Aldrich Co., Poznan, Poland), succinic anhydride (97%, Aldrich Co.), N,N′-dicyclohexylcarbodiimide (DCC, 98%, Aldrich Co.), triethylamine (TEA, ≥99%, Sigma-Aldrich Co.), and 4-(dimethylamino)pyridine (DMAP, ≥99%, Aldrich Co.) were used without further purification. Dichloromethane (DCM, anhydrous, ≥99.8%, POCh, Gliwice, Poland), dimethylsulfoxyde (DMSO, anhydrous, 99%, Aldrich Co.), tetrahydrofuran (THF, anhydrous, 99.8%, POCh, Gliwice, Poland), and diethyl ether (anhydrous, 99.8%, POCh, Gliwice, Poland) were used as received. Dimethylsulfoxyde-d_6_ (DMSO-d_6_) in ampoules for NMR measurements (99.9 atom% D) was purchased from ARMAR Chemicals (Döttingen, Switzerland). Phosphate buffer solution (pH 7.40 ± 0.05, 0.1 M, PBS, potassium dihydrogen phosphate/di-sodium hydrogen phosphate, 20 °C, Avantor Performance Materials, Gliwice, Poland) was also used as received.

### 2.2. The ROP Procedure

The biodegradable polymers attached to β-CD molecules were prepared using different molar ratios of the initiator (β-CD) of the monomers: d,l-lactide (LA), l,l-lactide (LLA), and ε-caprolactone (CL) (Table 1). The initiator/monomer feed ratio for the synthesized macromolecules ranged from 1/50 to 1/100. The obtained products were denoted as β-CD/PLA50, β-CD/PLA100, β-CD/PLLA50, β-CD/PLLA100, β-CD/PCL50, and β-CD/PCL100, where β-CD = β-cyclodextrin, PLA = poly(d,l-lactide), PLLA = poly(l,l-lactide), and PCL = poly(ε-caprolactone). For each polymerization, dry β-CD and CL, LA, or LLA were accurately weighted and introduced into 5-mL glass ampoules. The reactions were carried out under argon atmosphere in an oil bath thermostated at 100 or 140 °C for 48 h. Then, the reaction products were cooled down and dissolved in dry dimethylformamide (DMF). The resulting product solutions were poured into the excess of dry tetrahydrofuran (THF )to precipitate β-CD. The unreacted β-CD was collected by centrifugation. The remaining extracts were dried in vacuo. After that, a dry residue was dissolved in DCM, precipitated twice from cold diethyl ether, and dried under vacuum for 72 h. The resulting polymers were characterized by ^1^H NMR spectroscopy and ^13^C CP/MAS NMR in the solid state.

**Data for β-CD/PCL100:**^1^H NMR (DMSO-d_6_, 300 MHz; δH, ppm) [19]: 5.76–5.67 (m, –OH2 and –OH3 of β-CD), 4.83 (d, C-1H of β-CD), 4.44 (br, –OH6 of β-CD), 4.32 (br,–CH_2_OH, terminal moiety of PCL(**e**)), 3.98 (t, –CH_2_O– of PCL (**d**)), 3.64 (br, C-3,6H of β-CD), 3.55 (br, C-5H of β-CD), 3.38 (m, -CH_2_OH, terminal moiety of PCL (**d’**)), 3.31 (br, H_2_O, C-2,4H of β-CD), 2.27 (t, –CH_2_CH_2_C(O)– of PCL (**a**)), 2.20 (t, –CH_2_CH_2_C(O)– terminal moiety of PCL (**a’**)), 1.55 (m, –CH_2_CH_2_C(O)– of PCL (**b+b’**)), 1.30 (m, –CH_2_CH_2_CH_2_– of PCL (**c+c’**)); ^13^C CP/MAS NMR (100 MHz; δC, ppm): 173.7 (–CH_2_CH_2_C(O) (**a+a’** of PCL)), 103.9 (C-1 of β-CD), 99.0 (C-1′ of β-CD), 81.9 (C-4 of β-CD), 73.1 (C-2, 3, and 5 of β-CD), 65.4 (–CH_2_O– (**e** of PCL), 64.1 (–CH_2_OH (**e’** terminal moiety of PCL)), 61.2 (C-6 of β-CD), 33.3 (–CH_2_CH_2_C(O) (**b+b’** of PCL)), 29.2 (–CH_2_CH_2_C(O) (**c+c’** of PCL)), 25.8 (–CH_2_CH_2_CH_2_– (**d+d’** of PCL)).

**Data for β-CD/PLA100:**^1^H NMR (DMSO-d_6_, 300 MHz; δH, ppm): 5.20 (m, –CH(CH_3_)– of PLA (**a**)), 4.92 (q, -CH(CH_3_)– lactide unit of PLA attached to β-CD (**a’**)), 4.84 (br, C-1H of β-CD), 4.19 (q, -CH(OH), terminal moiety of PLA (**a’’**)), 3.64 (br, C-3,6H of β-CD), 3.55 (br, C-5H of β-CD), 3.40–3.10 (m, C-2,4H of β-CD), 1.46 (m, -CH_3_ of PLA (**b+b’)**), 1.28 (d, -CH_3_ terminal moiety of PLA (**b’’**)); ^13^C CP/MAS NMR (100 MHz; δC, ppm): 170.0 (-CHC(O) (**a+a’** of PLA)), 102.9 (C-1 of β-CD), 95.5 (C-1′ of β-CD), 81.8 (C-4 of β-CD), 72.8 (C-2, 3, and 5 of β-CD), 69.5 (-CH(CH_3_)– (**b+b’** of PLA), 60.5 (C-6 of β-CD), 20.5 (-CH(CH_3_)- (**c’** terminal moiety of PLA)), 16.8 (-CH(CH_3_)- (**c** of PLA).

### 2.3. Synthesis of the PDL Conjugates

The synthesized branched matrices −2 g of β-CD/PLA100, β-CD/PLLA100, and β-CD/PCL100 were first dissolved in 50 mL anhydrous THF under stirring for 30 min at room temperature. Then, succinic anhydride (the molar ratio of the polyester matrices to succinic anhydride was 1:1.3 for each hydroxyl group; see Table 1) and 1.5 mL of triethylamine (TEA) as the catalyst were added to the reaction flask under equal reaction conditions for 24 h. The crude product (carboxyl terminal groups) was then precipitated in the cold diethyl ether twice and dried in a vacuum (the products yield ranged from 84 to 85%). Subsequently, to a solution of the obtained products (1.5 g of each matrix) in DMSO (25 mL), *N*,*N*′-dicyclohexylcarbodiimide (DCC) and 4-dimethylaminopyridine (DMAP) were added under nitrogen (the molar ratio of polyester products with a carboxyl terminal group to DCC was 1:1.2 for each carboxyl group, and the molar ratio of DCC to DMAP was 1:1.5). The mixture was stirred for 2 h at room temperature, and PDL was added (the molar ratio of polyester matrices with a carboxyl terminal group to PDL was 1:1.5 for each carboxyl group). The reaction was stirred for 72 h at room temperature. The crude conjugation products had been precipitated three times in a cold diethyl ether and dried in a vacuum. The yields of the conjugation products were 70, 69, and 74% for β-CD/PLA100/PDL, β-CD/PLLA100/PDL, and β-CD/PCL100/PDL, respectively.

**^1^H NMR of β-CD/PCL100/PDL** (DMSO-d_6_, 300 MHz, δH, ppm): 11.06 (s, -NH of PDL (**L**)), 7.20 (s, -CH= of PDL (**K**)), 7.02 (m, -CH= of PDL (**I**+**H**), 6.47 (m, -CH= of PDL (**J**+**G**)), 5.79-5.63 (m, -OH2 and -OH3 of β-CD), 4.84 (d, C-1H of β-CD), 4.45 (br, -OH6 of β-CD), 3.99 (t, -CH_2_O- of PCL (**d+d`**) and -OCH-, -CH_2_- of PDL (**A**+**F**), 3.61-3.53 (br, C-3,5,6H of β-CD), 3.32 (br, H_2_O, C-2,4H of β-CD and -CH_2_-, -CH- of PDL (**B**+**D**), 2.81 (d, -CH_2_- of the linker (**e**), 2.27 (t, -CH_2_CH_2_C(O)– of PCL (**a+a`**)), 1.59–1.06 (m, –CH_2_CH_2_C(O)- of PCL (**b+b’**), -CH_2_CH_2_CH_2_– of PCL (**c+c’**) and -NH- of PDL (**C**), -CH_3_ of PDL (**E**).

**^1^H NMR of β-CD/PLA100/PDL** (DMSO-d_6_, 300 MHz, δH, ppm): 11.06 (s, -NH of PDL (**L**)), 7.20 (s, -CH= of PDL (**K**)), 7.02 (m, -CH= of PDL (**I**+**H**), 6.47 (m, -CH= of PDL (**J**+**G**)), 5.24-5.19 (m, –CH(CH_3_)– of PLA (**a+a`+a``**)), 4.82 (br, C-1H of β-CD), 4.42–4.00 (br, -OH6 of β-CD and -OCH-, -CH_2_- of PDL (**A**+**F**)), 3.54–3.10 (br, H_2_O, C-2,3,4,5,6H of β-CD and -CH_2_-, -CH- of PDL (**B**+**D**), 1.69–1.02 (m, –CH_3_ of PLA (**b+b`+b``**) and -NH- of PDL (**C**), -CH_3_ of PDL (**E**).

### 2.4. Molecular Simulations

#### 2.4.1. Structure Choice and Preparation

Molecular modeling calculations were performed using Accelrys’ Materials Studio^®^ (Accelrys, San Diego, CA, USA). β-CD/PCL molecule has been chosen as the model one for the molecular simulations. Multiple isomers, differing in degree of polymerization (*DP)* and degree of substitution (*DS)* as well as the position of substitution of the polymer chain to the β-CD macrocycle, were generated and compared in terms of their conformations and energies. Isolated polymer chains were constructed from the repeat units using the polymer builder function and used to substitute chosen hydroxyl groups of β-CD. Initially, the chains were constructed with random torsions between the repeat units. The initial structure of unsubstituted β-CD has been taken from Cambridge Crystallographic Data Centre, refcode BCDEXD10. The Forcite module (COMPASS force field, Smart algorithm with ultrafine quality and maximum 50,000 iterations) was then used to obtain the energy-minimized conformation of each isomer.

#### 2.4.2. Conformational Search

Conformational analysis of the studied isomers was performed using the Conformers module of Accelrys’ Materials Studio^®^, applying the Metropolis Monte Carlo algorithm based Boltzmann jump search method. The total number of 10^3^ conformers per isomer were generated, applying 500 perturbations per jump within the torsion angle window adjusted to accept 50% of the generated conformers. Geometry optimization has been performed on each conformer by the provided Smart algorithm (COMPASS force field, ultrafine quality, and maximum 50,000 iterations).

### 2.5. Toxicity Assays

#### 2.5.1. Microtox and Spirotox Tests

Five milligrams of the polymeric products was placed in the glass tube with 5 mL of Tyrode’s solution (Spirotox) or 2% NaCl (Microtox). The tubes were incubated at 37 °C for 24 h with shaking.

Microtox is a short-term bioassay with the luminescent bacteria *Aliivibrio fischeri* (previously known as *Vibrio fischeri*). The procedure was based on the International Organization for Standardization (ISO) standard [29]. Shortly, the tested and the control (2% NaCl) samples were incubated with the bacteria at 15 °C for 15 min, and the luminescence was measured in the Microtox M500 luminometer (ModernWater, London, UK). Then, the percent of inhibition of the luminescence was calculated in comparison to the control.

Spirotox is a short-term bioassay with the ciliated protozoan *Spirostomum ambiguum*. The test was performed according to standard protocol [30]. Shortly, the tested and the control (Tyrod solution) samples were incubated with the protozoans at 25 °C for 24 h, and the sublethal (deformations) and lethal effects were observed with the dissection microscope (magnification 10×). The percentage of affected protozoans for the sample was then determined as compared to the control.

#### 2.5.2. Umu-test

Umu*-*test detects the induction of the SOS system in the strain *Salmonella typhimurium* TA1535/pSK1002. SOS system is the bacterial response to the DNA-damaging agents. The test strain is genetically modified—the umuC gene activity is linked to the synthesis of β-galactosidase, while other DNA regions responsible for this enzyme synthesis were deleted. Therefore, β-galactosidase activity strictly depends on the SOS system: induction level and the genotoxic activity of the tested compound [31]. The enzyme converts the colorless substrate (ortho-nitrophenyl-β-galactoside) into the yellow product, which can be quantified colorimetrically at 420 nm. Additionally, the bacteria growth (G) is evaluated by measurement of an optical density to determine the toxicity of the tested samples. The genotoxic potential of the sample is presented as the Induction Ratio (IR)—the β-galactosidase activity ratio of the tested sample in comparison to the negative control. Samples with IR ≥ 1.5 are considered as genotoxic.

In the present study, the umu-test was carried out in the micro-plate variant according to the ISO 13829 guideline, with and without metabolic activation (S9 liver fraction from male Sprague–Dawley rats treated five days before the isolation with a single dose of 500 mg/kg body weight of Aroclor 1254 in soya oil). Deionized sterile water was used as a negative control, 2-aminoanthracene and 4-nitroquinoline *N*-oxide were used as positive controls, and phosphate-buffered saline (PBS from Gibco, Thermo Fisher Scientific, Darmstadt, Germany) was used as solvent control. All tested samples were incubated in PBS-1 mg/mL for 24 h, 37 °C, with shaking. Before the assay, all extracts were sterilized by filtration (0.20 µm). All samples were tested in two-fold dilution series (four concentrations, the highest concentration of 0.66 mg/mL).

### 2.6. PDL Release Study from the Branched Conjugates

The in vitro release study of PDL from the obtained branched conjugates β-CD/PLA100/PDL, β-CD/PLLA100/PDL, and β-CD/PCL100/PDL (in the powder form) was investigated by measuring the concentration of PDL released at pH 7.40 ± 0.05. All experiments were carried out in triplicate. Nine hundred mg of dried conjugates was immersed in 100 mL of buffer solution (pH 7.40 ± 0.05) and incubated at 37 °C, with continuous rotation at 50 (cycles/min). Ten mL of samples was collected from the release medium at fixed time intervals using the filter followed by replacing a fresh buffer solution with 10 mL. The absorption of buffer solution at the absorbance peak with a wavelength of 264 nm was measured by a UV-Vis spectrophotometer [32]. The absorbance peak was correlated very well with the concentration of PDL. A linear calibration curve was obtained by calculating the absorption of solutions with predetermined PDL concentrations. The absorbance readings were within the reference range for all measurements in this study.

### 2.7. Measurements

Conventional ^1^H NMR experiments were done at 298 K using a Varian VNMRS spectrometer (Palo Alto, CA, USA) operating at 300 MHz for ^1^H, using the standard Varian software. Proton spectra were acquired with 128 scans and the relaxation delay of 1 s. All samples were dissolved in DMSO-d_6_. The solid-state NMR measurements were carried out using a Bruker Avance WB-400 spectrometer (Banner Lane, Coventry, UK) with the resonance frequencies of 400 MHz for ^1^H and 100 MHz for ^13^C. For the ^13^C CP/MAS experiments, a 4-mm double-bearing CP/MAS probe was used. The proton decoupled spectra were collected at 298 K under MAS at 7.5 kHz, with the proton π/2 pulse of 2.40 μs, contact time of 8, and 20 ms, and the optimized recycle delay of 100 s the number of transients was 72.

Correlation 2D ROESY (Rotating frame nuclear Overhauser effect spectroscopy) measurements were carried out at 298 K on a Varian VNMRS spectrometer operating at 300 MHz for protons, using the Varian VnmrJ software. The NMR experimental conditions were as follows: 64–128 transients per increment for 256 increments, 300 ms mixing time, and 1024 complex data points in the F2 domain. The spectra were processed with Gaussian window functions in both dimensions.

Thermogravimetric analysis (TGA) was performed on SDT Q600 (V20.9 Build 20) instrument (TA Instruments, New Castle, Delaware, USA). Approximately 20 mg of sample was loaded in the platinum pan and heated at a constant heating rate of 5 °C min^−1^ from room temperature to 600 °C under argon atmosphere. The temperature of the polymer degradation was defined as the extrapolated onset of the lowest temperature weight loss.

MALDI-TOF mass spectroscopy measurements were performed on a Bruker UltraFlex MALDI TOF/TOF spectrometer (Bremen, Germany) in a linear mode using DHB (2,5-dihydroxybenzoic acid) or HABA (2-(4’-hydroxybenzeneazo) benzoic acid matrix and Bruker Peptide Calibration Standard (1047.19–3149.57 Da) as a calibrant and analysed with flex Analysis v.3.3 (Bruker Daltonik GmbH) and Polymerix v. 2.0 (Sierra Analytics Inc.) software (Modesto, CA, USA). The matrix was dissolved in THF or DCM at a concentration equal to 10 mg mL^−1^ and mixed with a polymer sample in a 25:1 *v*/*v* ratio.

Number-average molecular weights (*M*_n_) and polydispersity indexes (*M*_w_/*M*_n_) were measured using SEC-MALLS instrument composed of an 1100 Agilent isocratic pump, autosampler, degasser, thermostatic box for columns, a photometer MALLS DAWN EOS (Wyatt Technology Corporation, Santa Barbara, CA, USA), and differential refractometer Optilab Rex. ASTRA 4.90.07 software (Wyatt Technology Corporation, Santa Barbara, CA, USA) was used for data collecting and processing. Two 2x TSKgel MultiporeHXL columns (Aldrich Co.) were used for separation. The samples were injected as a solution in methylene chloride. The volume of the injection loop was 100 mL. Methylene chloride was used as a mobile phase at a flow rate of 0.8 mL min^−1^.

The amount of the released PDL was quantitatively determined by a UV-Vis spectrophotometry (UV 1202 Shimadzu, Duisburg, Germany) in aqueous buffered solutions at the adsorption maximum of the free drug (λ = 264 nm) using a 1-cm quartz cell.

## 3. Results and Discussion

### 3.1. Structural Characterization of the Matrices

The biodegradable and/or bioresorbable polymers were successfully obtained during ROP of cyclic esters using the β-CD as initiator of a natural origin. The synthesized polyester matrices can be further used as carriers for conjugation with drugs in order to receive effective materials for controlled drug delivery. All the polymeric samples were synthesized under anhydrous conditions. Table 1 summarized the results concerning ROP of CL, LA, and LLA to form β-CD/PCL, β-CD/PLA, or β-CD/PLLA polymeric products. The initiator to monomer ratios ranged from 1/50 to 1/100 at reaction time 48 h and reaction temperature 100 or 140 °C.

The synthesized products were characterized by means of ^1^H NMR and ^13^C CP/MAS NMR (Figure 1, Figure 2 and Figure 3). The β-CD/PCL100 polymer was also characterized by ^13^C NMR and heteronuclear single quantum correlation (HSQC) spectra (see Appendix A, Appendix A), whereas β-CD/PLA50 was characterized by rotating frame overhause effect spectroscopy (ROESY) spectra (see Appendix A, Appendix A). Figure 1 shows the ^1^H NMR spectrum of the β-CD/PCL100 polymeric sample. There are clearly visible internal methylene proton signals of PCL chain at 3.99, 2.27, 1.53, and 1.30 ppm, assigned as **d**, **a**, **b**, and **c**. Also, it could be observed the terminal methylene proton signals of PCL chain at 3.38 and 2.20 ppm, denoted **d’** and **a’**, whereas the terminal signals denoted **b’** and **c’** were overlapped with the internal ones assigned as **b** and **c**. The signal at 4.34 ppm was probably derived from the hydroxyl group of the terminal moiety of PCL and appeared in the presence of water. In addition, signals derived from β-CD aliphatic protons H-1, H-3, H-5, and H-6 and hydroxyl groups OH-2, OH-3, and OH-6 were also detected.

The structures of β-CD/PLA and β-CD/PLLA polymers were characterized by ^1^H NMR, as illustrated for the β-CD/PLA50 sample in Figure 2. The spectrum shows the typical proton signals of methine (**a**, 5.19 ppm) and methyl groups (**b**, 1.46 ppm) of the LA unit in the polymer chain. The methine protons signal from the terminal moiety (**a’’,** 4.20 ppm) was clearly visible as well as the terminal methyl (**b’’,** 1.29 ppm). Furthermore, the presence of the lactide unit attached directly to the initiator β-CD molecule gives methine **a’** a 4.97-ppm signal, whereas methyl **b’** was overlapped with the internal one. The signals derived from all β-CD aliphatic protons (H-1 to H-6) as well as the hydroxyl groups of OH-2, OH-3, and OH-6 were also well pronounced. A decrease in the area under the signals of the hydroxyl groups (OH-2 and OH-3) of the β-CD was observed, indicating the presence of β-CD initiator in the macromolecule (Figure 1 and Figure 2). There are potentially 21 hydroxyl initiating sites in the β-CD macrocycle. The calculations of the average degree of substitution (*DS*) for the β-CD/PCL polymeric samples were based on the area ratio of the terminal methylene proton signal **d’** (Figure 1) to the signal area of the β-CD H-1 proton at 4.83 ppm derived from the 7 glucose units. For the β-CD/PLA or β-CD/PLLA polymers, the calculations of *DS* were based on the area ratio of the terminal methine proton signal **a’’** (Figure 2) of PLA or PLLA to the signal area of the β-CD H-1 proton. The average degree of polymerization (*DP*) was calculated based on the area ratio of the terminal methylene proton signal **d’** (Figure 1) to the internal methylene proton signal **d** (Figure 1) for the β-CD/PCL polymeric samples, whereas the areas of the terminal methine proton signal **a’’** (Figure 2) of PLA or PLLA to the internal methine proton signal **a** (Figure 2) were compared for the β-CD/PLA or β-CD/PLLA polymers.

As is shown in Table 1, the *DS* of β-CD/PCL, β-CD/PLA, and β-CD/PLLA ranged from 11.3 to 19.6 for the polymeric samples obtained at 140 °C, meaning that approximately twenty initiating sites were initiated the growth of twenty polymer chains. Only for the sample β-CD/PCL50, the *DS* value was calculated to be 3.3. Significantly lower values of the *DS* (ranged from 8.0 to 8.7) were observed for the polymeric samples synthesized at 100 °C (Table 1). The calculated results clearly confirm the authors’ earlier assumptions about lower reactivity not only in LA and LLA monomers but also in the CL monomer at 100 °C.

Lower reactivity of cyclic esters at 100 °C was also confirmed by the calculated average molecular weight values (*M*_n(NMR)_) of the polymeric samples, determined by ^1^H NMR spectroscopy (Table 1). The average molecular weight of β-CD/PCL100 polymeric sample obtained at 140 °C is more than 2.5 times higher than those obtained at 100 °C (see Figure 3a). Similarly, *M*_n_ of β-CD/PLA100 sample obtained at 140 °C is two times higher than those obtained at 100 °C, whereas for β-CD/PLLA100 sample the increase is about 1.7 times. Importantly, the *M*_n(NMR)_ values were in a good agreement with the *M*_n(SEC-MALLS)_, determined using the SEC-MALLS method (Table 1). In addition, the size-exclusion chromatography (SEC) elution curve presented for the β-CD/PCL100 sample shows one mode (see Appendix A, Appendix A), indicating that the obtained polymeric product was monomodal with a low molar-mass dispersity (ÐM) value. Importantly, almost two times lower yield of the polymerization products, obtained under the temperature of 100 °C in comparison to 140 °C, confirms the author’s supposition about lower reactivity of the cyclic esters at 100 °C (see Figure 3b).

The solid-state NMR measurements are an excellent tool for studying solid structure of compounds containing cyclodextrins. ^13^C NMR shifts in solid of glucose units of native β-CD are very characteristic, and any changes of chemical shifts and appearing of new signals have provided very valuable information about the possible substitution in the glucose units. Therefore, the ^13^C CP/MAS NMR spectra of obtained β-CD/PCL100 and β-CD/PLA100 polymeric samples were analyzed and compared to the native β-CD (Figure 4). The spectrum of β-CD/PCL100 polymer (Figure 4b) shows the pronounced carbon-13 signals of the polymerized CL: carbonyl (**a**) and methylene (**b**,**c**,**d**,**e**). Also, the terminal methylene signal (**e’**, 64.1 ppm) from the terminal moiety of PCL was detected. The rest of the terminal signals overlapped with the internal ones.

The spectrum of β-CD/PLA100 polymer (Figure 4c) demonstrates the carbon-13 signals of the polymerized LA (**a**,**b**,**c**), carbonyl signal (**a’,** 173,5 ppm), and methyl signal (**c**’, 20.5 ppm) from the terminal unit of the PLA. In the ^13^C CP/MAS NMR spectra of β-CD/PCL100 and β-CD/PLA100 (Figure 4b,c), the significantly altered carbon-13 signals of the β-CD moiety can be distinguished in comparison to native β-CD (Figure 4a). Although the C-2 and C-3 signals (bond to the secondary hydroxyl groups) strongly overlapped and their changes were difficult to observe, the change in the position of the C-6 signal (connected to the primary hydroxyl groups) is significant and indicates the β-CD incorporation to the polymer. Furthermore, the observation of additional C-1′ signals, shifted upfield, confirms substitution of the polymer chain to the β-CD macrocycle at the position of 2 and/or 3. The C-1 signal is the most sensitive to the changes in torsion angle of the glycosidic linkage between glucose units [33,34]. Since there are modified and unmodified glucose units in the β-CD macrocycle substituted with the polymer chains, the torsion angles differ from one another and several C-1 and C-1′ signals were observed. Besides, the chemical shifts of C-2,3,5 signals in the spectra of β-CD/PCL100 and β-CD/PLA100 polymeric samples were altered compared to native β-CD. The thermogravimetric analysis (TGA) of polymers primarily provides information about thermal stability, and then, it allows to find out the decomposition mechanisms for the investigated materials [35]. TGA was carried out to investigate the influence of the β-CD incorporation on the thermal stability of the obtained polymers. To the best of authors’ knowledge, the PCL and PLA materials functionalized with β-CD have not been studied in terms of thermal stability so far. The comparison of the thermal degradation of the β-CD/PCL100 polymer, 100% PCL, and pure β-CD have been shown on Figure 5. The TGA curve of a pure β-CD shows 10% loss of water from the β-CD cavity at 75.6 °C and the two-step degradation: first the rapid one at 305.1 °C and the gradual between 320 and 520 °C. The TGA curves of β-CD/PCL100 polymer, however, show that the polymer degradation has slowly appeared near the temperature of β-CD degradation. Then, the mass loss occurred in a single step with the onset decomposition temperature of 369.3 °C for β-CD/PCL100 (100 °C) and 371.9 °C for β-CD/PCL100 (140 °C), which reflect the complete decomposition of the samples. Pure PCL starts to decompose at the temperature of 310.5 °C. It can be seen that the incorporation of β-CD molecule to the β-CD/PCL100 macromolecule increased thermal stability of the polymer compared to the PCL homopolymer, probably due to the branched structure of the polymeric material functionalized with β-CD. Until the β-CD was decomposed, the β-CD/PLA100 sample was stable. After the β-CD degradation, the PCL chains decompose at a higher temperature than PCL homopolymer. Abdolmohammadi and coworkers found the same behaviour for the degradation of the PCL/chitosan blend [36], where the composite PCL/chitosan had better thermal stability than a pure PCL. In their work, the first step of decomposition corresponded to the polysaccharide degradation resulting from the saccharide ring dehydration and the second one belongs to the PCL degradation.

For an additional confirmation of the presence of β-CD moiety in the macromolecule, the obtained polymeric materials were characterized by the matrix-assisted laser desorption/ionization time-of-flight mass spectrometry (MALDI-TOF MS). Figure 6 shows the MALDI-TOF mass spectrum of the β-CD/PCL100 polymer obtained at 100 °C. The mass of β-CD is 1134.98 Da, and a spacing of 114 *m/z* between two peaks has to be assigned to the mass of the repeating unit in the PCL. The spectrum includes four main series of peaks: first is characteristic of PCL with one β-CD end group and one hydrogen end group (R_M_ = 1157, Na^+^ adduct, A4); the second is also assigned to PCL with one β-CD end group and one hydrogen end group complexed with potassium counter-ions (R_M_ = 1173, B4), and the third and fourth are both characteristic of PCL with one hydroxyl end group and one hydrogen end group (C14, R_M_ = 18, Na^+^ adduct and D14, R_M_ = 18, K^+^ adduct). The presence of linear polymers (terminated with hydroxyl end group and hydrogen end group) could result from hydrolysis, which took place during the isolation of the polymers or during the preparation of the samples for the MALDI TOF-MS measurements. Likewise, the MALDI-TOF mass spectrum of the polymeric sample β-CD/PCL100 synthesized at higher temperature (140 °C) shows four main series of peaks, terminated in both cases with one β-CD end group and one hydrogen end group (A4 and B4) and the two following: with one hydroxyl end group and one hydrogen end group (C14 and D14) (see Appendix A, Appendix A).

However, the MALDI-TOF mass spectrum of β-CD/PLA100 shows two main series of peaks (Figure 7). In this case, both of them are corresponding to PLA terminated with β-CD end group and one hydrogen end group: first, A12, R_M_ = 1157, Na^+^ adduct and, second, complexed with potassium counter-ions (R_M_ = 1173). Similarly, the MALDI-TOF mass spectrum of β-CD/PLA100 obtained at 140 °C shows two main series of peaks (see Appendix A, Appendix A). In the mass spectrum in Figure 7, two populations of chains were detected, that is with even and odd numbers of LA repeating units. They were separated by 72 *m/z*. The two populations can be explained by the transesterification reaction, which is typical for the polymerization of LA [37].

### 3.2. Molecular Modeling Studies

To further explore the structural properties and the relative stability of the selected isomers of the β-CD/PCL50, molecular modeling methods have been applied. β-CD/PCL50 has been chosen as a model compound due to fact that, according to the ^1^H NMR and SEC-MALLS analysis results, it was the sample characterized by the lowest values of *DP* and *DS*. Therefore, it was interesting whether there will be any differences between the structures and stabilities based on the energy differences in the isomers differing in the type of substitution, as defined in Figure 8. It was also interesting whether the substitution occurs at C_2_ position, as suggested by Harada et al., [19,20,21,22] or, alternatively, at C_3_ position. To answer those questions, eight isomers of β-CD/PCL (*DP* = 5, *DS* = 4, C_162_H_270_O_75_) have been modeled, as described in Section 2.4. Further, the obtained isomers were compared with another two (*DP* = 20, *DS* = 1, C_162_H_270_O_75_), differing in the position of substitution in β-CD (OH-2 or OH-3). The obtained results have been summarized in Table 2.

Analysis of the data from Table 2 indicates the differences in the stability between the studied isomers. In both sets (OH-2 and OH-3 substitution), the isomers with larger *DS* were found to be more stable, which was in agreement with experimental results. In all of the ten studied isomers, the formation of the helical secondary structure was observed (see Appendix A, Appendix A). In both sets (OH-2 and OH-3 substitution), “Type B” was found to be the most stable one. This is probably caused by the steric effects and by the fact that, in this type, the substitution is the most evenly distributed, as only in this type there is solely one group of S-S, meaning two directly bonded glucopyranose units being substituted. This explanation is also confirmed by the fact “Type A” substitution, in which four successively connected glucopyranose units have been substituted, was found to possess the highest energy. However, since the isomers characterized by the higher *DS* were found to be the lower energy ones, it indicates that some beneficial, in terms of stability, van der Waals interactions between the polymeric chains occur. Further, for all of the studied types of substitution (A–D) as well as *DS* = 1, the substitution at the OH-2 position resulted in the formation of more stable polymers, which is in agreement with Harada’s results [19,20,21,22]. However, since the energy differences between the OH-2- and OH-3-substituted isomers were not found to be very large, in comparison to the differences between the A–D types of substitution, it suggests that both OH-2 and OH-3 substitutions may occur, which is in agreement with the experimental (^1^H NMR) results from this work. The energetic preference of the “B” type of substitution may have a direct impact on the release profile and release efficiency. It should be noticed that, in order to release the PDL from the conjugate, a hydrolysis must occur. Therefore, the ability of the water molecules to freely access the PDL is crucial for this reaction. In the “B” substitution, the PDL molecules are distributed most homogeneously, which may prevent the formation of local supersaturation and thus affect the release process.

### 3.3. Toxicity Studies

In the Spirotox and Microtox tests, it was found that none of the tested samples had been toxic to the protozoan *S. ambiguum* and luminescent bacteria *A. fischeri* (Table 3).

The same findings have been identified in the umu-test with *S. typhimurium;* all of the tested samples showed no toxic and no genotoxic potential with or without metabolic activation (IR < 1.5, Table 4).

The obtained results are not equal with a classic evaluation of the clinical safety and are presented here only as an additional data on the influence of tested samples on the living cells. At this stage of our research, we are unable to predict the final forms of medical devices or implants that may contain materials that are the main topic of this report. If all of these details were clarified, all the required tests will be carried out.

### 3.4. The PDL Conjugates’ Synthesis and Drug-Release Characteristics

Polymer-drug conjugates have been actively designed as potential DDSs also for β-adrenolytics. We have therefore focused our work on the synthesis (Scheme 1) and characterization of novel and branched β-CD/biodegradable polyester/PDL conjugates for local delivery.

By a simple esterification reaction, the molecules of PDL were successfully conjugated to the β-CD/PCL100, β-CD/PLA100, and β-CD/PLLA100 backbones, which were denoted as β-CD/PCL100/PDL, β-CD/PLA100/PDL, and β-CD/PLLA100/PDL.

As we reported on Figure 1, the hydroxyl protons of the terminal moiety of PCL presented at 4.32 ppm were not observed on Figure 9, confirming conjugation of PDL at free hydroxyl groups available on the carrier units. Meanwhile, the protons related to the indole ring of PDL were apparent in the regions of 11.06, 7.20, 7.01, and 6.49 ppm (see Appendix A, Appendix A
^1^H NMR spectra of a pure PDL). The protons corresponding to the aliphatic chain of the β-blocker molecule were also observed; however, they have been overlapped with methylene proton signals of the PCL chain. It is also worth to note that the proton signal of a free hydroxyl group of PDL moiety (5.03 ppm, **M**; see Appendix A, Appendix A) was not observed in the β-CD/PCL100/PDL spectrum, indicating functionalization of the primary hydroxyl group of the active substance. The internal methylene proton signals of PCL chain as well signals derived from β-CD aliphatic protons H-1, H-3, H-5, and H-6 and hydroxyl groups OH-2, OH-3, and OH-6 were still observed in Figure 9.

Analysis of the β-CD/PLA100/PDL conjugate spectrum could create similar conclusions (see Appendix A, Appendix A).

The number of PDL molecules conjugated to the branched carrier can be calculated from NMR spectrum based on the -NH-proton of the indole ring of PDL (**L**) at 11.06 ppm and the aliphatic proton **1** of β-CD at 4.82 ppm as shown in Figure 9 and Appendix A. The results were found to be 10 and 8, which implies that 10 and 8 PDL molecules were conjugated to the β-CD/PCL100 and β-CD/PLA100 carriers, respectively.

The hydrolytic release of PDL from the branched carriers was investigated in phosphate buffer solution (PBS), pH 7.40 ± 0.05, over 20 days. The results are graphically demonstrated in Figure 10.

The PDL release from all β-CD/biodegradable polyester/PDL conjugates was considerably slow over 20 days, showing efficient conjugation of the drug to the polymer chain with slow hydrolytic degradation. As depicted in Figure 10, about 18, 29, and 41% of the covalently conjugated PDL were released from β-CD/PCL100/PDL, β-CD/PLLA100/PDL, and β-CD/PLA100/PDL samples for 24 h. However, the cumulative drug release from the synthesized materials increased up to 32, 44, and 57% after 20 days, respectively. Meanwhile, no “burst release” was observed in all three synthesized samples. The results indicated that the designed conjugates offered an impactful approach for the sustainable release of β-blocker under physiological conditions.

## 4. Conclusions

Our study demonstrated successful polymerization of cyclic esters in the presence of β-CD as a natural, operationally simple, inexpensive, and safe-for-human-health initiator of ROP. The polymerizations were performed with no solvents and without introducing any metal impurities. The monomers ε-caprolactone, d,l-lactide, and l,l-lactide were selected due to their many desirable properties for medical applications. Besides, the combination of these monomers with β-CD exhibited inefficient polymerization results in the previously reported studies. The branched polymeric materials terminated with β-CD were successfully obtained and characterized. The spectroscopic data clearly indicate the incorporation of β-CD molecule into polymer chain. The presence of β-CD signals in both ^1^H NMR and ^13^C CP/MAS NMR spectra, in particular the changes in β-CD carbon-13 chemical shifts of signals C-1 to C-1’ and C-6, confirmed the substitution of the polymer chain to the β-CD ring. Compared to previous studies, where β-CD/PCL100 carrier was obtained with a yield of 15% and the number-averaged molecular weight of 1700 Da, our work has produced the same material with a yield of 91% and an average molecular weight of 17,000 Da. Furthermore, TGA analysis has shown that the incorporation of β-CD to the β-CD/PCL100 polymer increases thermal stability of the polymer compared to the PCL homopolymer. The resulting matrices were also subjected to toxicity and genotoxicity assays, demonstrating their nontoxic and nongenotoxic potential with or without metabolic activation. The molecular modeling methods have been applied to further investigate the structural properties and the relative stability of the selected isomers of the β-CD/biodegradable polyester sample. On the basis of this approach, we have tried to estimate whether there will be any differences between the structures and stabilities, based on the energy differences, in the isomers that differ in the substitution type. Importantly, novel branched conjugates of PDL for a local delivery have been synthesized and thoroughly characterized in our study. The in vitro release studies showed that, with these branched matrices, the duration of drug release can be prolonged (by more than 20 days) with no “burst release”. The results indicated that the conjugate design offered an effective approach to the sustainable release of β-blocker under physiological conditions. However, further improvement in potency and greater specificity of conjugates may be needed for this type of polymer-drug conjugate, and we are continuing to investigate these potential improvements.

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
