# Peer review of "Development and Evaluation of Matrices Composed of β-cyclodextrin and Biodegradable Polyesters in the Controlled Delivery of Pindolol"

_pharmaceutics, 2020, doi:10.3390/pharmaceutics12060500_

Round 1

Reviewer 1 Report

The presented paper is devoted to the synthesis of polyesters with application of cylodextrin as initiator. Similar studies have been published previously and the novelty of obtained materials is unclear. Authors didn't review and discuss the following paper in the introduction: Qiang Zhang et al. Cyclodextrin-centred star polymers synthesized via a combination of thiol-ene click and ring opening polymerization. Chem. Commun., 2012, 48, 8063–8065. There also other similar research, which was not addressed, when describing the objectives of the study.

The paper contains very poor pharmaceutical soundness and seem better fitted to the journal with materials/polymerization scope. I really wonder why the authors decided to send this manuscript to Pharmaceutics.

Comments:

  1. The introduction is badly written, and the objectives of the work are not clear. The authors start with arterial hypertension treatment significance, but further Lines 60-114 contain too much information from polymer science area. So there is no connection between these parts. The only discussed studies are ref 26 (the thesis from Cyclodextrin Symposium in 2000) about transcorneal permeability and ref 28. However, authors didn’t explain - what their biological targets of treatment are? By the way, I wonder why authors cite the old Symposium data, while there is a recent paper on similar topic: Isabel Rodríguez et al Enhancement and inhibition effects on the corneal permeability of timolol maleate: Polymers, cyclodextrins and chelating agents, International Journal of Pharmaceutics 529 (2017) 168–177? Thus the pharmaceutical soundness of introduction is very poor. The authors didn’t explained the proposed formulation preparation strategy and the way of application? The relation between formulation application and possible biological effects are not addressed.
  2. The results and discussion part doesn’t contain the enough relevant discussion on motivation for many experiments to be done. There are too much NMR descriptions of materials, which were already described previously in numerous studies. However, the formation of desired cyclodextrines with grafted chains of polyesters is not well-proved. The 1H NMR with diffusion filter, DOSY NMR and COSY NMR techniques are required (at least one of these methods) to prove the formation of covalent bonds, but not just of the mixture. Authors have used SEC to determine the MWs of the products. So it would be useful to show the SEC elution curve to show how many modes there are in the system.
  3. Line 383. The “conception” word is too pathos here. It seems to be just the outcome of thermodynamic laws, which are already described for such systems in Handbook of Ring-Opening Polymerization.
  4. Line 398. The motivation to use 13C CP/MAS NMR is unclear. This should be explained.
  5. Line 443. The TGA doesn’t prove the formation of new chemical unit. This could be similar for the mixture as well. The DSC measurements will be valuable. It would be interesting to address the crystallinity variation in pure PLLA, PDLLA and PCL, in mixture with CD and in branched CD-polyesters structures. The crystallinity is very important factor for release from such materials. The materials morphology studies by SEM or AFM are also of importance.
  6. Line 476. The molecular modeling studies are interesting, but irrelevant to the pharmaceutical research.
  7. Line 512. The methodology and motivation for such biological testing is unclear. The results are not supported by discussion. In my opinion, such testing is irrelevant to the prepared formulation.
  8. Line 529. The Scheme is not at the right place.
  9. Line 525. The cyclodextrines application in drug delivery formulations is usually related to inclusion of the drug into the CD ring. However, authors decided to make the conjugates by using the polyesters terminal carboxylic group. The need for CDs in that case is unclear. The formation of conjugate is not supported by diffusion related NMR methods. Just the presence of corresponding peaks could not be enough for such justification.
  10. Line 553. What are the Drug Loading and Encapsulation efficacy parameters? In which form was the material for the release studies (it is NOT described also in Experimental part, Line 278). Films, powder,…? What was the surface area? The Fig. 10 shows obvious burst release, revealing the fact that there is conjugated and non-conjugated drug in the system. It is obvious, that just hydrolysis of ester bonds could not give the 20, 30 and 40 % growth of cumulative release within 1-2 hours. The free drug is definitely present in the system.  However, authors declare NO burst release, which is a mistake. The authors didn’t tried to make the release modeling to analyze the mechanism of release from their systems. The natural medium of the organism (blood, ECM) contains lipases and esterases, which will accelerate the ester bond cleavage. Thus the choice of just neutral PBS as the medium for release studies is strongly wrong.  
  11. The English spelling is definitely requires careful revision: lines 18-20 in the abstract; lines 54-55; lines 73-74 (vacuo/vacuum); line 590 (“branched matrices” – are the polymers branched, or the matrices). These are just examples from wide list.

Author Response

Thank you. 

Reviewer 2 Report

Dear Editor,

the manuscipt entitled "Development and evaluation of matrices composed of β-cyclodextrin and biodegradable polyesters in the controlled delivery of pindolol" regards the synthesis of polymer-drug conjugates for the treatment of hypertension. Although the synthesis of the polymeric products is not new, the different conditions used by the authors demostrate how it is possible to improve the yield of the products. Characterization of the products by means of NMR and other techniques is exhaustive and clearly supports the conclusions drawn by the authors. The drug release is also very interesting and it could be promising for the release of other drugs. The manuscript is well written and all results are clearly explained.

I recommend to accept the paper with minor revision.

Please provide my detailed comments in the attached pdf file.

Author Response

Thank you. 

Reviewer 3 Report

The manuscript submitted by Oledzka and co-workers is full of results, however I have some doubts about its suitability for publication. This work deals in the first part with the synthesis of polyester conjugated with cyclodestrins. Then the authors propose a method of further conjugation with a bioactive compound, the pindolol, and finally they evaluated the release of pindolol under certain conditions.

One of the weakest point of the entire manuscript, in my opinion, is the release experiment of the pindolol. This is also the most important part of the work because it deals with the possible pharmaceutical applications. How did the authors design this experiment? There is no chance of an ester hydrolysis in a buffer solution at 7.4 pH. A mechanism should be highlighted and discussed or I have missed something! In my opinion, the authors should better discuss this part and should furnish more accurate evidences about the analytical method indicating the presence of the only pindolol in solution.

I understand the conjugated polymer is not soluble in water solution, otherwise this would interfears with release experiments. However, this is also a very weak point for pharmaceutical applications, the solubility. Usually, these release systems are based on water soluble polymers like PEG, hyaluronic acids. This point should be better discussed.

In addition, it is not clear what is the advantage with respect to literature synthesis of similar CD-polyester conjugates as reported by Harada and others.

In general I think that the manuscript is too long, especially the introduction part. Too many acronyms, a list of them would help.     

Author Response

Thank you. 

Reviewer 4 Report

Some minor comments:

Line 47 : Avoid the repetition of "All of these drugs"

Line 117 : could you add some name of marketed products (Visken, Visken Quize, ..)

Line 157 : replace by dimethylsulfoxyde

Line 159 : replace by dimethylsulfoxide-d6

Line 165 : add the corresponding hydroxyl/monomer ratio

Line 245 : replace by Aliivibrio fischeri

Table 1 : replace Mw/Mn by molar-mass dispersity ĐM.

Line 456 : the presence of linear polymers could be the result of the presence of water during the polymerization.

Line 476. From the 1H NMR spectra, the authors could determine the number by CD of OH-6 and OH-2/OH-6 substituted but no information was find in the manuscript.

Line 563. “Meanwhile, no “burst release” was observed in all three synthesized samples.” The reviewer is not totally agree with the authors. Perhaps the authors can show an enlarger part of the beginning of the release profiles.

Author Response

Thank you. 

Round 2

Reviewer 1 Report

Dear Authors,

In my honest opinion the paper published in Pharmaceutics should contain information, which is valuable for pharmacological society and scientist involved in such research. The presented paper is very interesting and valuable from the point of view of macromolecular chemistry, but do not contain very important information for the system under investigation to be applied in pharmacology.

The authors have made some corrections and improved the “chemical” part of the paper. However, the part devoted to pharmaceutical application stayed practically unchanged. Most of my comments (#1, 2, 5, 6, 7, 9, 10, 11) are answered as personal comments from authors to me, but didn’t involve the manuscript text corrections. 

I’ll comment some replies of the authors and give my principal comments and opinion further.

Comment #1.

The authors have appealed to the scope of Pharmaceutics, which is : ”Covered topics include pharmaceutical formulation, process development, drug delivery, pharmacokinetics, biopharmaceutics, pharmacogenetics, and interdisciplinary research involving, but not limited to, engineering, biomedical sciences, and cell biology.” However, they did not mention to which topic they attribute their work. If those are “pharmaceutical formulation” and “drug delivery”, the possible pharmacological benefit over previous formulations should be stated or possible effect of new formulation on biological properties. Lines 141-142 in introduction: “Nevertheless, according to the literature and our knowledge, branched conjugates of PDL constructed from biodegradable polyester chains and β-CD, which form the central core, have not yet been obtained and described. In this sense, the rate of drug release from the synthesized conjugates has been analyzed in vitro.” There is no statement on possible interest from pharmacological society.

Comment #2.

The SEC elution curve is not for me, but it will be useful for readers. It should be in Supplementary information.

Comment #3

The authors wrote in their answer: “Obviously, we agree with the Reviewer that it would be interesting to explore the aspect of crystallinity of our samples using DSC analysis. Currently, the studies of the production of these systems on a larger scale are underway. Full thermoanalytical analysis and the adjustment the drug release profiles to the mathematical models will also be performed. Such a set of studies will allow us to fully discuss the mechanism of the carrier degradation and to establish a full correlation between the products' average molecular weight, their crystallinity, etc. and the kinetics of pindolol released also in vivo. These data will be published in our next work.” I wonder how one can discuss the release without knowledge of drug loading, encapsulation efficacy, crystallinity of the polymer, pore structure of the polymer, polymer degradation data, surface area and so on. It seems to me that authors have no control over the release from proposed materials and the results are rather occasional than systemic. Thus I would suggest to authors to publish their current results in the journal with “polymeric scope” and after they do the proposed experiments they could present them in pharmaceutical journals. The results of branched polymer synthesis and it’s study by NMR are very interesting for me, but I’m really not sure about Pharmaceutics readership.

Comment #6

I have wrote: “The molecular modeling studies are interesting, but irrelevant to the pharmaceutical research.”

Authors have answered: “Thank you for agreeing with us that the molecular modeling studies presented in this article are indeed interesting. To justify their pharmaceutical relevance we would like to give some arguments. First of all, molecular modeling methods have found their widespread applications in the pharmaceutical studies i.e. to explain and support the experimental results at the molecular level, such in the submitted study, but also in the other studies already published in the MDPI Pharmaceutics and other pharmaceutical journals. Some examples….”

Of course, I know that modelling is a powerful tool in pharmacology and I also have applied it in my own research. In this sense, the reply of the authors is redundant.

The main point here, that authors did nothing for making it understandable – how their own modelling study of “differences between the structures and stabilities, based on the energy differences, in the isomers differing in the type of substitution” is relevant to the only “pharmacological” property, which they provide? I mean release.  

First sentence in the paragraph: “To further explore the structural properties and the relative stability of the selected isomers of the β-CD/PCL50, molecular modeling methods have been applied. β-CD/PCL50 has been chosen as  a model compound due to fact that, according to the 1H NMR and SEC-MALLS analysis results, it was the sample characterized by the lowest values of DP and DS” and the last one from the section: “However, since the energy differences between the OH-2 and OH-3 substituted isomers were not found to be very large, in  comparison to the differences between the A-D types of substitution, it suggests that both OH-2 and  OH-3 substitutions may occur, which is in agreement with the experimental (1H NMR) results from this work.”, are not referred to the desired pharmacological properties of the material.

It is definitely should be stated - why this results are important in this study?! How this structural peculiarities affect the release?

Comment #7

Authors: “The umu-test is a bioassay to assess the genotoxic potential of various chemical compounds.”

Please explain in the text of the paper – why genotoxicity is important in this research? I still don’t understand the relevance of this text to the application of your systems.

Authors: “These tests were not supported by discussion because the obtained results were negative (no genotoxicity or toxicity detected).”

In my honest opinion any result, positive or negative, should be mentioned and discussed somehow in the text?

Comment #10

I have asked about many properties of materials “What are the Drug Loading and Encapsulation efficacy parameters? In which form was the material for the release studies (it is NOT described also in Experimental part, Line 278). Films, powder,…? What was the surface area?”

The authors have met only one of those: “We have added the information about the form of the obtained materials for the release studies (please see line 281).”

The sentence at line 280-282 is following: “The in vitro release study of PDL from the obtained branched conjugates β-CD/PLA100/PDL, 280 β-CD/PLLA100/PDL and β-CD/PCL100/PDL (in the powder form) was investigated by measuring 281 the concentration of PDL released at pH 7.40±0.05.”

So the authors gave only information on the form of material - powder. Was the size of particles in the powder small or large? Is there any polydispersity of the particles within the powder? Was the specific surface area measured to analyze the area of contact between two phases?

Please explain – how you can observe 20-40 % of release within first several hours if all drug was conjugated covalently and no free drug in the system? Is it possible for covalent bond to cleave so fast in enzyme-free medium?

Authors state in the in conclusions (lines 605-607): “The in vitro release studies showed that with these branched matrices, the duration of drug release can be prolonged (by more than 20 days) with no “burst release”. The results indicated that the conjugate design offered an effective approach to the sustainable release of β-blocker under physiological conditions.” However, one can observe “burst” release and just PBS could not be used as “physiological conditions”.

My principal comments:

  1. The effect of polymer branched structure on the crystallinity and morphology of obtained formulation prototypes should be shown.
  2. The particles should be characterized by SEM, DLS or AFM in order to evaluate their size and distribution.
  3. The drug loading capacity and efficacy should be provided.
  4. The drug release mechanism should be analyzed by mathematical modelling. What governs release – diffusion of dissolution? Compare the release in PBS and release with enzymes in the medium (the second one will be much more rapid and will not take so much time).
  5. The question should be answered – if the release drug amount allowed the pindolol concentration to fall into the “therapeutic window” during all time of release?

Best regards,

           Reviewer.

Author Response

Kindly please see the attachment.

Reviewer 3 Report

The manuscript has been improved and, therefore, can be accepted for publication. The authors gave a detailed list of answers, which have been greatly appreciated.

Best regards

Author Response

We thank the Reviewer for his/her kind assessment of our work.  

Round 3

Reviewer 1 Report

The principal comments were not met by the authors.

No major revision was made, but just formal reply provided.

1.The authors comment, that the cannot calculate the "Encapsulation efficacy", because it is conjugate. So they should provide the data on "conjuagtion efficancy". How much of the drug was taken for conjugation and how much of that was conjuagted? The drug loading means - how much of the drug was in the formulation, which was taken for drug release. This information was not provided by the authors. 

The authors have argued as follows: "Regarding the drug loading: as we have described in our work “The number of PDL molecules conjugated to the branched carrier can be calculated from NMR spectrum based on the -NH- proton of the indole ring of PDL (L) at 11.06 ppm and the aliphatic proton 1 of β-CD at 4.82 ppm as shown in Figures 9 and S810. The results were found to be 10 and 8, which implies that 10 and 8 PDL molecules were conjugated to the β-CD/PCL100 and β-CD/PLA100 carrier, respectively” (please see lines 562-566). Based on the above data, we can estimate the molar drug content in the macromolecule that we made and presented as "cumulative drug release" by "%" (see Figure 10)." 

However, this phrase does not provide the clarification for the drug loading. It is still unclear - how much of the drug was in the formulation before the release measurement was started? The authors should calculate and provide these data.

2. The authors have argued on drug release data: "We have clearly demonstrated in the paper that by a simple esterification reaction, the molecules of PDL were successfully conjugated to the β-CD/PCL100, β-CD/PLA100 and β-CD/PLLA100 backbones (lines 553-555). In our opinion, the hydrolysis of ester bonds occurs in acidic, alkaline, neutral environments and in the presence of enzymes."

Form the text of the paper I can only conclude that your release medium was just PBS. It is in the caption to Fig. 10. So it is not clear how you can have 20-40 % of release in  around 2 hours! This should be clarified and discussed.

My question on the "burst release" was not answered by the authors. 

3. The comment of authors "The data provided in the reviewed article are just preliminary findings that we are planning to expand. As a result, currently, the studies on the production of these systems on a larger scale are underway. Full physicochemical analysis and the adjustment the drug release profiles to the mathematical models, will also be performed. The release data points will be subjected to zero-order and first-order kinetics, and Higuchi or Korsmeyer–Peppas models to evaluate the kinetics and release mechanisms of the drug from the obtained carriers. Such a set of studies will allow us to fully discuss the mechanism of the carrier degradation and to establish a full correlation between the products' average molecular weight, their crystallinity, etc. and the kinetics of pindolol released, also in vivo. These data will be published in our next work." 

It will take just 1-2 days to apply you release data with mentioned release models to give some information on drug release mechanism. In my opinion this should be done for this publication to be published in Pharmaceutics.

4. Line 544 "If all of these details were clarified, all the required tests will be carried out." the phrase looks very strange and I think it should be modified or deleted.

5. The form, in which the formulation was applied for release, should be better described. Particles size and disstribution. Crystallinity and porosity.

Best regards,

Reviewer.